# Correlation Analysis between AK1 mRNA Expression and Inosine Monophosphate Deposition in Jingyuan Chickens

**DOI:** 10.3390/ani10030439

**Published:** 2020-03-06

**Authors:** Juan Zhang, Honghong Hu, Tong Mu, Weizhen Wang, Baojun Yu, Ju Guo, Ying Wang, Zihang Zhou, Yaling Gu, Zengwen Huang, Zhengyun Cai, Guosheng Xin

**Affiliations:** 1School of Agriculture, Ningxia University, Yinchuan 750021, Ningxia, China; zhangjkathy@126.com (J.Z.); h918622@163.com (H.H.); mtlv3h@163.com (T.M.); nxuwangweizhen@126.com (W.W.); yubaojunb@163.com (B.Y.); guoju_x@163.com (J.G.); wangyingnd@163.com (Y.W.); zhouzihang33@163.com (Z.Z.); guyl@nxu.edu.cn (Y.G.); xndaxue@126.com (Z.H.); caizhy@nxu.edu.cn (Z.C.); 2School of Life Science, Ningxia University, Yinchuan 750021, Ningxia, China

**Keywords:** Jingyuan chicken, inosine monophosphate, *AK1*, real-time PCR, mRNA

## Abstract

**Simple Summary:**

China is rich in livestock and poultry resources, but the development of animal husbandry in different regions is uneven, and excellent livestock and poultry resources have to be developed. Jingyuan chicken is a national-level protected breed of livestock and poultry and a genetically protected breed in the Ningxia Hui Autonomous Region. It is rich in amino acids and fatty acids, with a high nutritional value. It is the first choice of “green nutrition and health care” in chicken food. Meat flavor is the main factor affecting meat quality. Meat flavor is characterized by umami and aroma properties. Inosine acid (inosincacid, inosinemonphosphate, IMP) has the greatest influence on meat flavor and is an important indicator of the freshness of meat. The purpose of this study was to screen out differentially expressed genes related to IMP content in different parts of the Ningxia local breed of Jingyuan chicken on the basis of transcriptome sequencing, and select adenylate kinase 1 (*AK1*) for quantitative fluorescence verification and Pearson correlation analysis. These findings provide a theoretical basis for further discovery and screening of IMP-specific deposition regulation mechanisms, as well as scientific basis for the development and utilization of local varieties and the development of new approaches to molecular breeding.

**Abstract:**

In this study, we examined correlations between the deposition of inosine monophosphate (IMP) and mRNA expression of the adenylate kinase 1 (*AK1*) gene in Jingyuan chicken. The IMP content was determined by high-performance liquid chromatography. Transcriptome sequencing was used to screen the differentially expressed gene *AK1* and real-time quantitative polymerase chain reaction (PCR) to determine the expression level of *AK1* mRNA associated with IMP synthesis. IMP and inosine content in the breast muscles of both Jingyuan cocks and hens was found to be significantly higher than that in the leg muscles. Similarly, the expression of *AK1* mRNA in the breast muscles of cocks and hens was significantly higher than that in the leg muscles. Moreover, *AK1* mRNA expression in cock breast muscles was negatively correlated with IMP content, whereas its expression in cock leg muscles was positively correlated with IMP content. In contrast, the expression of *AK1* mRNA in hen breast and leg muscles was significantly positively correlated with IMP content. These findings provide a scientific basis for enhancing the meat flavor of Jingyuan chicken and promoting the development and utilization of local variety resources, as well as constituting a basis for screening IMP-regulated genes. Our study will advance our current understanding of *AK1* function.

## 1. Introduction

Chickens are among the most important sources of meat consumed by humans. Although in recent years, factors such as the avian influenza epidemic and Sino–U.S. trade war have had adverse effects, global chicken production and consumption continues to rise steadily. Compared with beef, lamb, pork, and other meat varieties, chicken and its products are growing in popularity, owing to their low price, short production cycle, ease of processing and storage, and perceived health benefits. Concomitant with this increasing popularity, there has also been an increase in demand for high-quality sustainably produced meat as living standards have improved in many regions of the world; consequently, to meet this growing demand, poultry breeders have sought to enhance the quality and flavor of the meat produced by avian livestock [1]. However, the quality of meat cannot be readily equated with a single index. Meat quality can be assessed in terms of a range of physical indices, including pH value, hydraulic strength, intermuscular fat content, shearing force, tenderness, flesh color, cooking loss, cooked meat rate, and muscle fiber diameter [2]. Among these, muscle pH can directly reflect muscle acidity, which in turn will directly influence properties such as muscle tenderness, drip loss, and flesh color [3].

Meat flavor is characterized by aroma and umami properties [4,5,6], which are primarily determined by volatile flavor compounds produced by the muscle matrix during the heating process. The umami flavor is derived from metabolites such as free amino acids and nucleotides, and particularly sodium inosine. Inosine monophosphate (IMP) is also a relatively important umami substance in livestock and poultry meat, which is formed by the deamination of adenosine monophosphate (AMP) under the action of adenosine monophosphate deaminase, and a higher content of inosine in chicken meat is generally associated with a more desirable taste. Water-holding capacity also has a significant influence on the physical properties and sensory characteristics of meat [7,8]. The IMP content has been used as one of the indicators to evaluate the freshness of meat [9,10], and to date, numerous studies have examined muscle IMP content in chickens, pigs, fish, and other livestock. Muscle IMP content is known to have a moderate level of heritability, and can also be affected by multiple factors, including genetic composition, nutrient levels, feed composition, slaughter methods, and storage conditions [11,12,13,14]. IMP is produced via two processes in vivo, namely, de novo and remedial syntheses; the former is notably complex, involving 10 steps catalyzed by 10 key enzymes. The expression levels of a range of enzymes have an important effect on determining the synthesis and metabolism of IMP.

Adenylate kinase (AK), a monomeric enzyme commonly found in both animals and plants, regulates adenine nucleotide metabolism and plays an important role in maintaining cell energy balance. Catalytic reaction: ATP + AMP ↔ 2ADP [15], and its activity has recently been found to be closely associated with apoptosis. Three subtypes of the AK gene have been identified, and in mammals, these different AK subtypes have tissue-specific distribution and unique subcellular localization [16]. In chickens, *AK1* is approximately 6 kbp in size and comprises seven exons [17]. It is considered to be the major AK subtype in the cytoplasm of skeletal muscle cells [18] and has been proposed as regulating extracellular adenine nucleotide pools by regulating receptors in skeletal muscle cells, thereby controlling cell signaling [19].

The Jingyuan chicken is an excellent egg-and-feather chicken breed native to China, which is characterized by a high tolerance to cold and arid climate of the Loess Plateau. It is mainly distributed in Jingning County in Gansu Province and Pengyang County in Ningxia Province. In 2006, it was recognized as an excellent local livestock and poultry breed by the Chinese Ministry of Agriculture. The breed has white, hemp-colored, and black feathered varieties, which have been designated national-level protected species of livestock and poultry genetic resources [20]. Owing to its superior environmental tolerance and unique feeding methods, the Jingyuan chicken is characterized by a high-quality meat that is rich in amino acids and fatty acids, with high nutritional value and good tonicity. Among chickens, it is considered a prominent “green nutrition and health care food” [21]. Therefore, we chose Jingyuan chicken as the research material, and our research group has also performed some research on the fatty acid composition, fatty acid-related genes, meat quality-related genes, and IMP-related genes of Jingyuan chicken in the early stage.

In this study, we sought to elucidate correlations between the IMP content in different parts of the local Ningxia cultivar of Jingyuan chicken and mRNA expression of *AK1*, on the basis of the results of previous transcriptome sequencing, and to verify the effect of *AK1* expression on the IMP content and meat quality. Thereby, our study provides a reference and theoretical basis to further research on the mechanism of IMP deposition with respect to *AK1* expression in Jingyuan chickens.

## 2. Materials and Methods

### 2.1. Experiment Material

The Jingyuan chickens used in this study were obtained from the Zhaona Chicken Breeding Center in Pengyang County, Ningxia. After maintenance under the same feeding and management conditions, 30 chickens (15 males and 15 females) were slaughtered at 180 days of age. Samples of the leg and breast muscles were rapidly collected, placed in a liquid nitrogen tank, and transferred to the laboratory, where they were stored at −80 °C and −20 °C for subsequent transcriptome sequencing, inosine and IMP content determination, and RNA extraction.

### 2.2. Determination of Inosine and IMP Content

Samples (10 g) of chicken breast and leg muscles were processed using a meat shredder, and 2 g aliquots of the resultant preparations were placed in 15 mL centrifuge tubes, to which 5 mL of perchloric acid was added followed by centrifugation at 8000 rpm for 10 min (repeated twice). Supernatants were collected in 100 mL volumetric bottles, and after adjusting the pH to 6.5, the volume was made up to 100 mL. The sample was then filtered through a 1-cm-diameter filter screen, and the filtrate was collected for high-performance liquid chromatography analysis [22].

### 2.3. Screening of the Differential Expression of AK1

The original data of transcriptome sequencing are the early data of the experiment. We selected the chest muscle with high IMP content and the leg muscle with low IMP content of Jingyuan chicken as samples for transcriptome sequencing. The sequencing platform was novoseq pe150. After the quality inspection of the total RNA of the extracted samples, the cDNA library was constructed, and the library was preliminarily quantified by Qubit 2.0. The library was diluted to 1ng (insert size), and then the inserted fragment length (insert size) of the library was detected by Agilent 2100. The Q-PCR method was used to accurately quantify the effective concentration of the library (the effective concentration of the library > 2 nM) to ensure the quality of the cDNA library. After the library test was qualified, HiSeq sequencing was carried out. After filtering the sequencing data, HISAT software was used to compare and analyze the reference genes of the filtered sequence, and HTSeq software was used to analyze the gene expression level of each sample. The fragments per kilobase of transcript per million fragments mapped (FPKM) values of the gene were subsequently calculated. Raw data of RNA-Seq was submitted on Sequence ReadArchive(SRA) in National Center for Biotechnology Information(NCBI), and the BioProject accession is PRJNA608986, whereas the BioSample accession is SAMN14217609. A comparison of differences in gene expression between breast and leg muscles revealed 2098 significantly differentially expressed genes, of which 1146 genes were significantly upregulated and 952 genes were significantly downregulated. *AK1* was screened out from the purine metabolic pathway via the Kyoto Encyclopedia of Genes and Genomes (KEGG) pathway analysis.

### 2.4. Protein Network Analysis of AK1

To predict whether *AK1* is related to genes in the inosine synthesis process, we performed a protein interaction network analysis of *AK1* using the STRING database.

### 2.5. Extraction and Reverse Transcription of Total RNA

The total RNA of Jingyuan chicken breast and leg muscles was extracted with TRIzol (Invitrogen, Carlsbad, CA, USA), and the purity, integrity, and concentration of the extracted RNA were subsequently determined. Reverse transcription was performed using the QuantiNova SYBR Green PCR kit and the synthesized cDNA was stored at −20 °C.

### 2.6. Primer Design

Primers were designed based on the GenBank sequences of Gallus *AK1* and *β-actin* using primer 5.0 software (Table 1) and sent to the Sangon Company for synthesis, using *β-actin* as the housekeeping gene.

### 2.7. Real-Time Fluorescence Quantitative PCR

Real-time PCR amplification was performed using the QuantiNova SYBR Green PCR kit (Table 2) in a total reaction volume of 20 μL containing the following reagents: 2 × SYBR Green PCR Master Mix, 10 μL; forward primer, 1 μL; reverse primer, 1 μL; RNase-free water, 7 μL; and template cDNA (100 ng/μL), 1 μL.

### 2.8. Data Processing

Two-way ANOVA of inosine and IMP content and gene expression was performed using SAS8.0 software. The relative expression level of the PCR target gene was statistically analyzed using the 2^−ΔΔ^Cycle threshold (CT) method [23]. The results are presented as the mean ± standard error. The correlation between IMP content and *AK1* mRNA expression was assessed using the bivariate correlation analysis.

## 3. Results

### 3.1. Determination of Inosine and IMP Content

Two-way ANOVA of the muscle inosine and IMP content in Jingyuan chickens revealed that under the same feeding and management conditions, the content of both IMP and inosine in the breast muscle of Jingyuan cocks was significantly higher than that in the leg muscles (*p* < 0.05 and *p* < 0.01, respectively). Similarly, in hens, the IMP and inosine content in the breast muscle was significantly higher than that in the leg muscle (*p* < 0.01). The comparison of IMP content between different sexes of Jingyuan chicken showed that the IMP content of Jingyuan hen was higher than that of cock, and that the content of inosine was lower than that of cock, but the difference was not significant. The interaction between different parts and sex of Jingyuan chicken was not significant. (Table 3).

### 3.2. Screening of the AK1 Gene in Transcriptome Data

#### 3.2.1. RNA-Seq Correlation Analysis

The results of our correlation analysis of differences in the gene expression levels in Jingyuan chicken breast and leg, between transcriptome sequencing samples are shown in Figure 1. The *R^2^* values of biological duplicates were ≥0.8, and the expression patterns between samples were highly similar. The principal component analysis of the samples showed in Figure 2 that breast-muscle and leg-muscle groups had great differences and good repeatability. Accordingly, we considered that the transcriptome sequencing results were reliable and could be used for the subsequent analysis.

#### 3.2.2. *AK1* Gene Screening

On the basis of the KEGG pathway analysis of differentially expressed genes screened from the transcriptome data (Table 4, *AK1* is marked with *), we identified three genes in the IMP synthesis pathway (purine metabolic pathway) showing differential expression, namely, *PKM2*, *AK1*, and *PGM1* (Figure 3), marked in red box.

#### 3.2.3. Gene Ontology (GO) Enrichment Analysis of *AK1*

The GO enrichment analysis of the screened *AK1* gene revealed that *AK1* is involved in 35 molecular functions and 12 biological processes, although it does not appear to be associated with any cell component-related process (Table 5).

#### 3.2.4. Protein Network Analysis of *AK1*

The results of our interaction network analysis of the differentially expressed *AK1* gene are depicted in Figure 4. The circles (nodes) shown in the figure represent the differentially expressed protein. Circles containing illustrations indicate that the gene has an associated protein structure, whereas empty circles indicate that the protein structures have not been determined for the gene. The figure shows that *AK1* and protein interact with 20 other genes, notably the two major enzymes ADSL and AMPD1 involved in purine synthesis.

### 3.3. qRT-PCR Amplification Product Specificity Analysis

Figure 5 shows the profiles of the total RNA extracted from the breast and leg muscles of Jingyuan chickens, which was separated electrophoretically on a 1% agarose gel. The three bands of 28S, 18S, and 5S RNA were clearly observed with no obvious indication of degradation. The OD_260_/OD_280_ values for the extracted RNA, determined using a Nanodrop spectrophotometer, ranged from 1.8 to 2.2, and the samples showed good integrity (an RNA integrity number (RIN) value ≥6.5), thereby indicating that they could be used for the subsequent analyses. Figure 6 shows the results of qRT-PCR amplification of the *AK1* target gene and *β-actin* reference gene, both of which showed s-type amplification curves with peaks between 18 and 30 Ct, high amplification efficiency, and good sample repeatability. The melting curves for both genes presented sharp and narrow peaks. No peaks indicative of non-specific fragments were detected between 60 and 70 °C, thereby confirming the good amplification specificity.

### 3.4. mRNA Expression of the AK1 Gene in the Breast and Leg Muscles of Jingyuan Chickens

Figure 7 shows that the expression levels of *AK1* mRNA in the breast and leg muscles were similar in cocks and hens, and that the expression in the breast muscle was significantly higher than that in the leg muscle.

### 3.5. Correlation between AK1 mRNA Expression and IMP Content in the Breast and Leg Muscles

The results of the correlation analysis between *AK1* mRNA expression and IMP content in the breast and leg muscles of cock and hen of Jingyuan chickens are presented in Table 6. The scatter plot with regression line is shown in Figure 8. We found that the *AK1* mRNA expression in the breast muscles of cocks was negatively correlated with the IMP content, whereas that in the leg muscles was positively correlated. In hens, the *AK1* mRNA expression was positively correlated with IMP content in both breast and leg muscles *(p* < 0.05).

## 4. Discussion

The demand for high-quality meat has been increasing in recent years. However, the quality of meat cannot be readily be equated with a single index. The types and content of flavor-related substances can differ considerably depending on multiple factors, including variety, gender, feeding methods, and different parts of a bird. Moreover, tastes relating to meat quality tend to differ with region, and accordingly, more extensive and in-depth research is needed to determine the factors that meet the needs and tastes of different individuals.

To this end, in the present study, we examined factors that potentially contribute to determining the quality of meat of the Ningxia local variety of the Jingyuan chicken breed. Initially, we determined the content of inosine and IMP in the breast and leg muscles of cock and hen birds. The results showed that the content of IMP in the breast muscle of Jingyuan chicken was 1.32 mg/g, which is relatively low compared with that of other native Chinese breeds such as Luoyang Wu chicken (2.57 mg/g), Liangfeng flower chicken (1.97 mg/g) [24], Luding chicken (2.65 mg/g), Miyi chicken (3.18 mg/g), and grass chicken (2.62 mg/g) [25]. However, it is more similar to the content in Wuding chicken (1.54 mg/g), Daweishan micro-chicken (1.34 mg/g) [26], Aibayi chicken (1.35 mg/g), and Beijing oil chicken (1.64 mg/g) [27]. In addition to differences in variety, we suspect that the observed differences in IMP content may be associated with differences in the age of birds, as well as differences in the methods used to preserve and treat samples. Moreover, it has been found that the expression of key enzymes involved in inosinic acid synthesis also differs at different developmental stages of different breeds [28,29]. In the present study, we also observed that IMP content in the breast muscles of cock and hen of Jingyuan chickens were significantly higher than that in the leg muscles, which is consistent with findings of previous studies [30,31,32]. It was found that the IMP content of Jingyuan hens was higher than that of Jingyuan cocks, whereas the inosine content of Jingyuan hens was lower than that of Jingyuan cocks but the difference was not significant, which was consistent with the previous research results [32,33,34]. This difference may be due to the differences in physiological conditions and metabolic intensity between male and female chickens.

On the basis of transcriptome sequencing of the breast and leg muscles of Jingyuan chicken, we identified three genes related to the de novo synthesis of IMP in the purine metabolic pathway that were differentially expressed. Among these, the Gene Ontology (GO) enrichment analysis indicated that *AK1* is associated with 35 molecular functions and 12 biological processes, whereas it showed no apparent association with cell component processes, which contrasts with the cytosolic GO results for *AK1* based on the STRING database. This is probably because we screened a different database for transcriptome sequencing. Analysis of the protein network interaction of *AK1* revealed interactive relationships with two pivotal genes involved in IMP synthesis, namely, *ADSL* and *AMPD1*—this indicates that *AK1* may be an influencing gene during IMP synthesis and has similar functions to ADSL and AMPD1. The results provide a theoretical basis to use *AK1* as a representative functional gene affecting IMP synthesis in the later experiments of our group.

AK1, a subtype of adenylate kinase, is a small cytoplasmic enzyme expressed in high-turnover cells, such as those in the skeletal muscle, blood cells, and brain, and its activity is dependent on the presence of Mg^2+^ or Mn^2+^ ions. *AK1* is considered the main AK subtype in the cytoplasm of skeletal muscle cells [18], and although some studies have shown that the muscle formation in *AK1*-deficient mice is normal, the abnormal accumulation of ADP in such mice has been found to be related to a delay in skeletal muscle relaxation [35]. *AK1* has been demonstrated to play a role in regulating adenine nucleotide metabolism, and in its absence, the amount of adenine nucleotides would be significantly reduced [36]. In addition, *AK1* phosphorylation may be necessary to transmit signals between plasma membrane mitochondria and K^+^ ATP channels [37]. In a study of Nanhua and Yorkshire pigs, Li et al. [38] identified 12 differentially expressed genes in the longissimus dorsi muscle by RNA-Seq; among the upregulated genes was *AK1*, and the subsequent fluorescence quantitative verification indicated that *AK1* is an important gene affecting meat quality traits. In the present study, the transcriptome sequencing results revealed that the expression of *AK1* mRNA in male and female chickens was significantly higher in the breast muscle than in the leg muscle. This is consistent with the results of RNA-Seq, thereby indicating that the RNA-Seq data were accurate and reliable. The correlation between *AK1* gene mRNA expression and IMP content in breast muscle and leg muscle of Jingyuan chicken showed that there was a significant positive correlation between *AK1* gene mRNA expression and IMP content in breast muscle and leg muscle of hen, and the correlation coefficients were 0.534 and 0.538, respectively, whereas the mRNA expression of *AK1* gene in chest muscle of cock was negatively correlated with IMP content, and the correlation coefficient was -0.404. The mRNA expression of *AK1* gene in leg muscle was positively correlated with IMP content, the correlation coefficient was 0.271, and the correlation degree was not significant. In addition, this study showed that the mRNA expression of *AK1* gene in chest muscle of cock was negatively correlated with IMP content, which was not consistent with that of hen and leg muscle, and was not consistent with the results of Dou Tengfei on inosinic acid content and ADSL gene expression of Wuding chicken and Daweishan miniature chicken. This may have been due to the small sample size and individual differences, and thus the sample size should be expanded for verification. *AK1* is a differential gene related to IMP, which has a certain effect on meat quality.

## 5. Conclusions

To the best of our knowledge, in this study, we discussed the differences of IMP mechanism in Jingyuan chicken due to different parts and sexes. Through real-time quantitative PCR and correlation analysis, we found the accuracy of screening differentially expressed genes by transcriptome sequencing to be high. Successfully screened *AK1*, PKM2, PGM1, and *AK1* promoted the formation of inosine acid in Jingyuan chicken; however, the specific function and mechanism of *AK1* is unclear. Therefore, it is necessary to conduct big data association analysis and functional research in the future. Screening of the Jingyuan chicken inosinic acid synthesis-related gene can be used as a marker for future marker-assisted selection in breeding.

## Figures and Tables

**Figure 1 animals-10-00439-f001:**
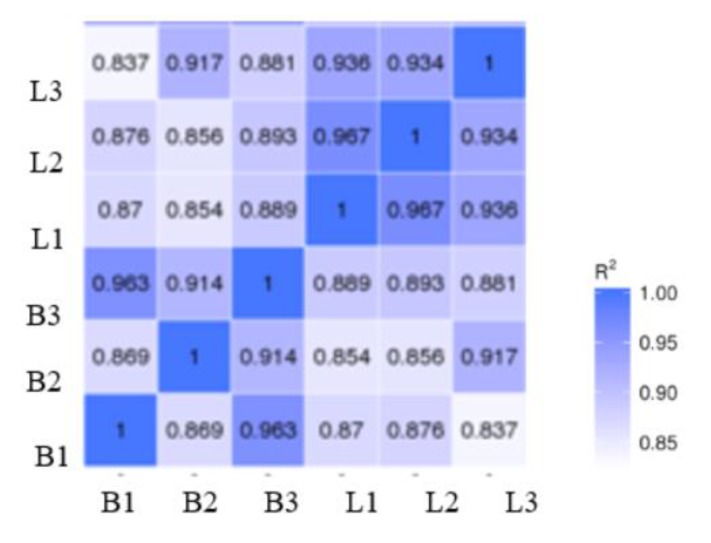
Sample-level correlation between transcriptome samples. Note: In the figure, B represents the breast muscle and L represents the leg muscle; *R^2^*: the square of Pearson’s correlation coefficient.

**Figure 2 animals-10-00439-f002:**
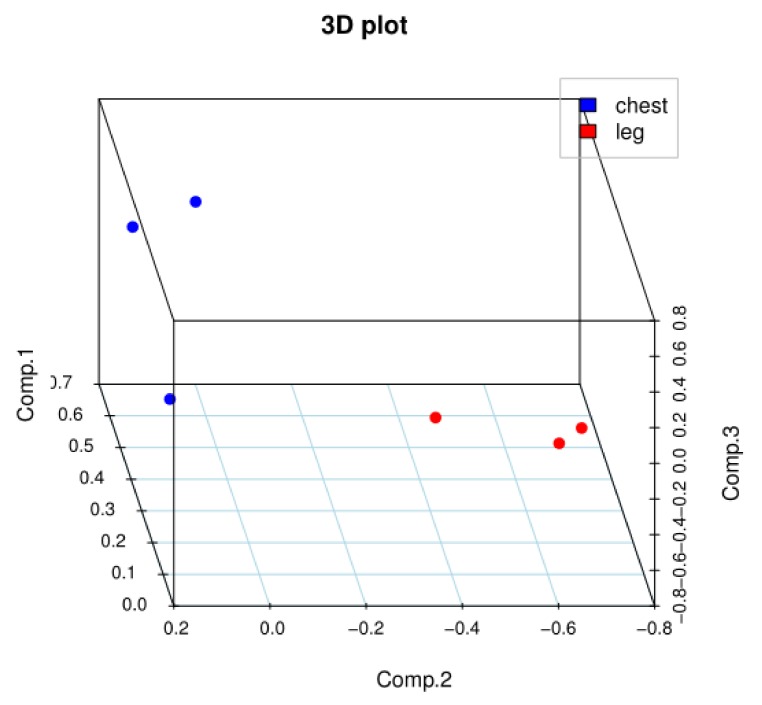
Principal component analysis of chest and leg.

**Figure 3 animals-10-00439-f003:**
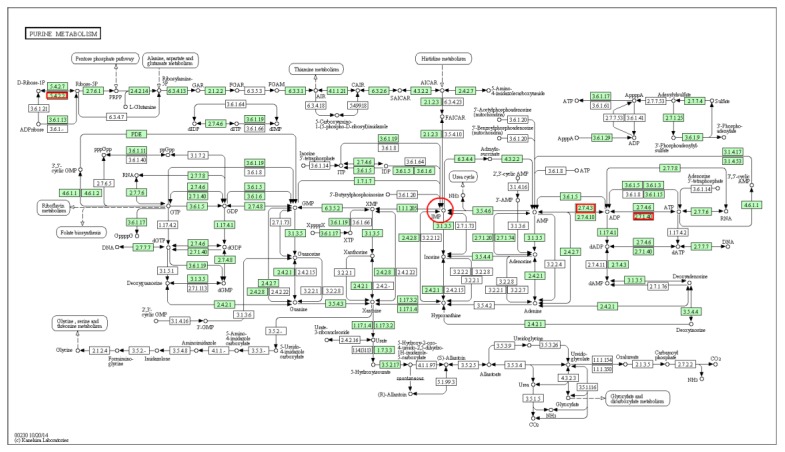
Location of differentially expressed genes (indicated by red boxes) in the purine metabolic pathway.

**Figure 4 animals-10-00439-f004:**
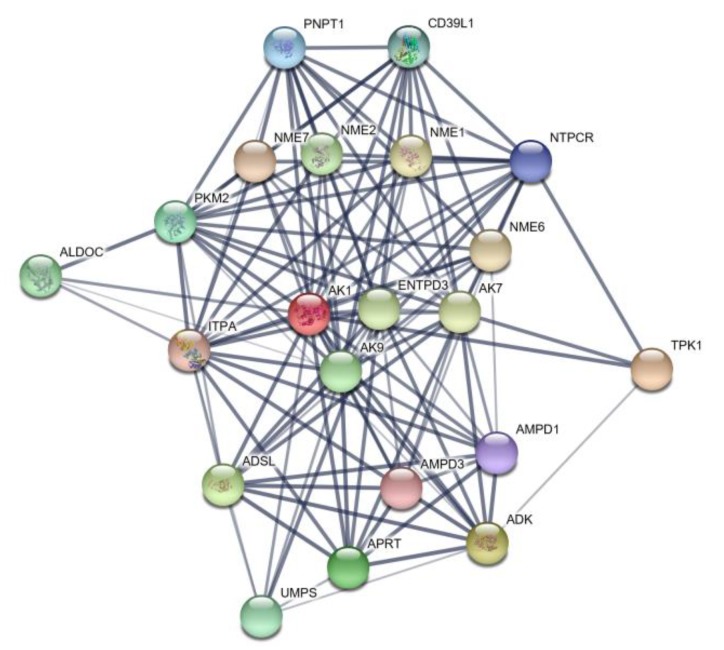
Protein network map of *AK1*.

**Figure 5 animals-10-00439-f005:**
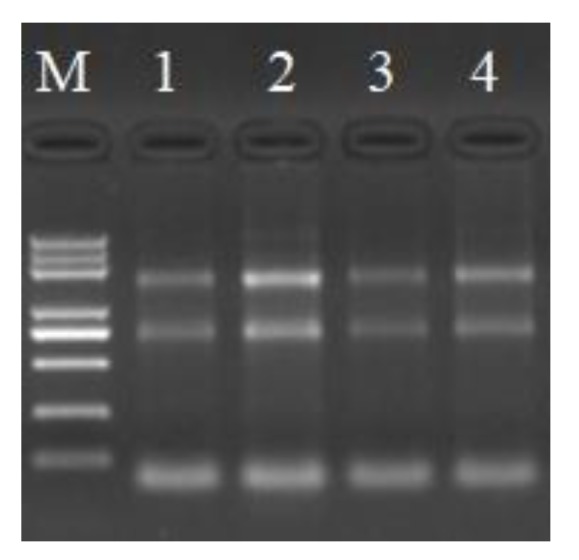
Total RNA extracted from different tissues of Jingyuan chicken. Note: M indicates Trans 2K Plus markers, 1–4 were diluted three times from the stock solution, and 1 µL was loaded; 1 and 2 represent the breast muscles, and 3 and 4 represent the leg muscles.

**Figure 6 animals-10-00439-f006:**
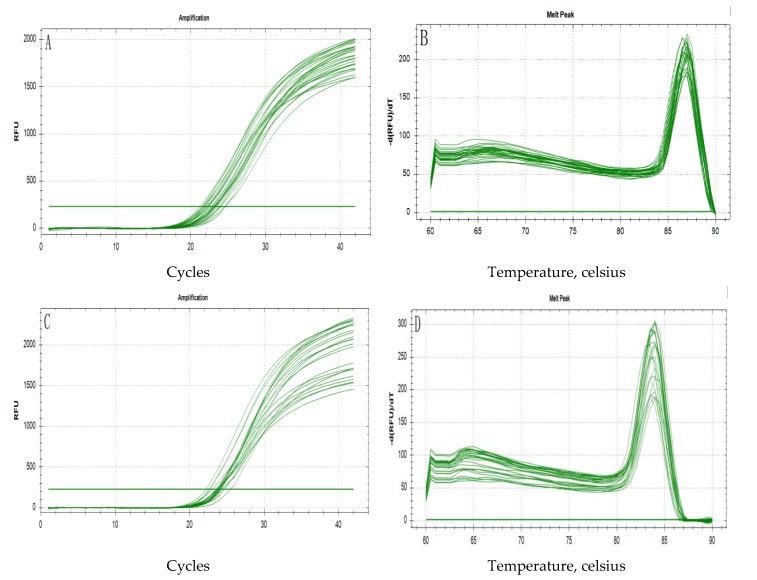
Amplification and melting curves based on the qRT-PCR analysis of *AK1* and *β-actin* RNAs. (**A**): Amplification curve for *AK1*; (**B**): dissociation curve for *AK1*; (**C**): amplification curve for *β-actin*; (**D**): dissociation curve for *β-actin*.

**Figure 7 animals-10-00439-f007:**
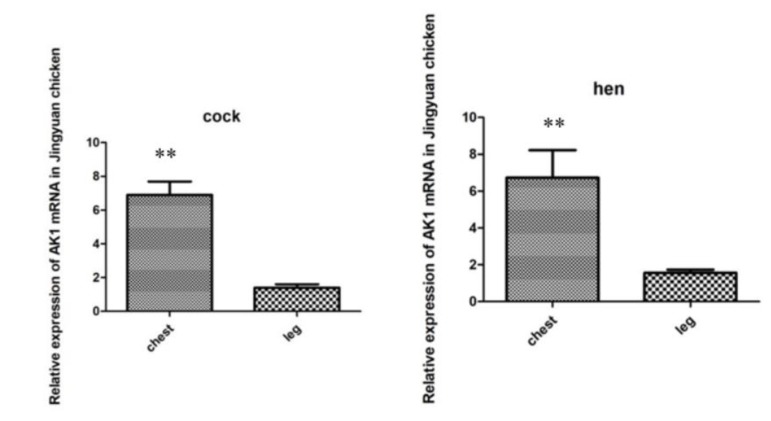
Comparison of *AK1* mRNA expression levels in the breast and leg muscles of cock and hen of Jingyuan chickens. ** represents significant difference in expression of AK1 in chest and leg muscles.

**Figure 8 animals-10-00439-f008:**
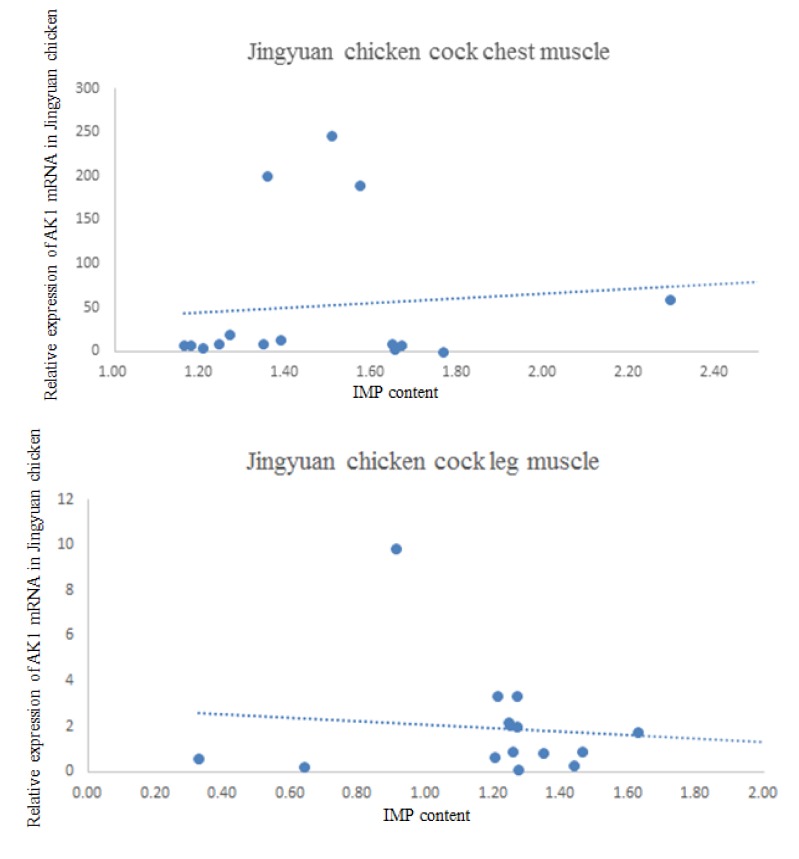
Jingyuan chicken *AK1* mRNA and IMP content scatter plot.

**Table 1 animals-10-00439-t001:** Primer sequences of adenylate kinase (*AK*)*1* and *β-actin* of Jingyuan chicken.

Gene	Accession Number	Primer Sequence (5′-3′)	Product Length (bp)
*AK1*	M37901	F: GGCTACCGAACCCGTCAT	185
		R: TAAGGCCACTTCCGCTGT	
*β-actin*	L08165	F: ATATTGCTGCGCTCGTTGT	191
		R: CAGGGTCAGGATACCTCTTTT	

**Table 2 animals-10-00439-t002:** Fluorescence quantitative PCR conditions.

Step	Time	Temperature	Ramp Rate
PCR initial heat activation	2 min	95 °C	Maximal/fast mode
Denaturation	5 s	95 °C	Maximal/fast mode
Combined annealing/extension	10 s	56.9 °C	Maximal/fast mode
Number of cycles	42		
Melting curve	0.05 s	60 to 90 °C	Increment 0.5 °C

**Table 3 animals-10-00439-t003:** Determination of inosine and inosine monophosphate (IMP) content.

Gender	Tissue	IMP Content (g/kg)	Inosine Content (g/kg)
Cock	Breast	1.484 ± 0.298 ^a^	0.359 ± 0.079 ^A^
	Leg	1.179 ± 0.329 ^b^	0.181 ± 0.096 ^B^
	Leg + breast	1.287 ± 0.247	0.270 ± 0.066
Hen	Breast	1.525 ± 0.294 ^A^	0.327 ± 0.067 ^A^
	Leg	1.200 ± 0.213 ^B^	0.181 ± 0.063 ^B^
	Leg + breast	1.362 ± 0.157	0.254 ± 0.035

Note: Different uppercase letters in the same column indicate an extremely significant difference (*p* < 0.01), whereas different lowercase letters indicate a significant difference (*p* < 0.05).

**Table 4 animals-10-00439-t004:** Differentially expressed genes in the purine metabolic pathway.

Term	Sample Number	Background Number	*p*-Value	Corrected *p*-Value	UniGenes	KO
Purine metabolism	3	148	0.769096085	0.993035152	ENSGALG00000001992 ENSGALG00000011016 ENSGALG00000029150 *	gga:396456 gga:424691 gga:396002 *

Note: * represents the selected gene number.

**Table 5 animals-10-00439-t005:** Results of the gene ontology (GO) annotation analysis.

Molecular Function (GO)						
GO Accession	Description	Term Type	Over-Represented	Corrected	DEG Item	DEG List
			*p*-Value	*p*-Value		
GO:0016301	kinase activity	molecular_function	0.000506	0.16646	145	1683
GO:0043167	ion binding	molecular_function	0.004572	0.46911	564	1683
GO:0003824	catalytic activity	molecular_function	0.004688	0.46911	755	1683
GO:0016787	hydrolase activity	molecular_function	0.009348	0.59761	370	1683
GO:0043168	anion binding	molecular_function	0.010458	0.62516	273	1683
GO:0016772	transferase activity, transferring phosphorus-containing groups	molecular_function	0.033078	0.91083	182	1683
GO:0001882	nucleoside binding	molecular_function	0.035078	0.91083	226	1683
GO:0016740	transferase activity	molecular_function	0.042141	0.97629	312	1683
GO:0035639	purine ribonucleoside triphosphate binding	molecular_function	0.043127	0.97629	224	1683
GO:0001883	purine nucleoside binding	molecular_function	0.045535	0.98066	224	1683
GO:0032549	ribonucleoside binding	molecular_function	0.045535	0.98066	224	1683
GO:0032550	purine ribonucleoside binding	molecular_function	0.045535	0.98066	224	1683
GO:0032555	purine ribonucleotide binding	molecular_function	0.048628	0.98066	224	1683
GO:0000166	nucleotide binding	molecular_function	0.049591	0.98066	256	1683
GO:1901265	nucleoside phosphate binding	molecular_function	0.049591	0.98066	256	1683
GO:0017076	purine nucleotide binding	molecular_function	0.052362	0.99569	226	1683
GO:0036094	small molecule binding	molecular_function	0.052961	0.99569	260	1683
GO:0032553	ribonucleotide binding	molecular_function	0.061359	1	225	1683
GO:0005524	ATP binding	molecular_function	0.086411	1	194	1683
GO:0032559	adenyl ribonucleotide binding	molecular_function	0.09077	1	194	1683
GO:0030554	adenyl nucleotide binding	molecular_function	0.095447	1	196	1683
GO:0005488	binding	molecular_function	0.10168	1	1139	1683
GO:0097367	carbohydrate derivative binding	molecular_function	0.10857	1	232	1683
GO:0016817	hydrolase activity, acting on acid anhydrides	molecular_function	0.13552	1	117	1683
GO:0097159	organic cyclic compound binding	molecular_function	0.22511	1	516	1683
GO:1901363	heterocyclic compound binding	molecular_function	0.22511	1	516	1683
GO:0016462	pyrophosphatase activity	molecular_function	0.31183	1	97	1683
GO:0043531	ADP binding	molecular_function	0.33714	1	6	1683
GO:0003674	molecular function	molecular_function	0.3384	1	1509	1683
GO:0017111	nucleoside-triphosphatase activity	molecular_function	0.34628	1	95	1683
GO:0016818	hydrolase activity, acting on acid anhydrides, in	molecular_function	0.39208	1	99	1683
	phosphorus-containing anhydrides					
GO:0004127	cytidylate kinase activity	molecular_function	0.57225	1	2	1683
GO:0019205	nucleobase-containing compound kinase activity	molecular_function	0.57356	1	8	1683
GO:0019201	nucleotide kinase activity	molecular_function	0.59085	1	6	1683
GO:0016887	ATPase activity	molecular_function	0.86048	1	38	1683
**Biological Process (GO)**						
GO:0044238	primary metabolic process	biological_process	0.0046851	0.46911	696	1683
GO:0071704	organic substance metabolic process	biological_process	0.0075438	0.56007	723	1683
GO:0008152	metabolic process	biological_process	0.034676	0.91083	828	1683
GO:0044237	cellular metabolic process	biological_process	0.047786	0.98066	652	1683
GO:0046483	heterocycle metabolic process	biological_process	0.18179	1	392	1683
GO:1901360	organic cyclic compound metabolic process	biological_process	0.20756	1	395	1683
GO:0006725	cellular aromatic compound metabolic process	biological_process	0.26552	1	387	1683
GO:0034641	cellular nitrogen compound metabolic process	biological_process	0.30598	1	424	1683
GO:0006139	nucleobase-containing compound metabolic process	biological_process	0.31487	1	368	1683
GO:0006807	nitrogen compound metabolic process	biological_process	0.32063	1	447	1683
GO:0008150	biological process	biological_process	0.54104	1	1216	1683
GO:0009987	cellular process	biological_process	0.76378	1	937	1683

**Table 6 animals-10-00439-t006:** Analysis of the correlation between IMP content and *AK1* mRNA expression in the breast and leg muscles of Jingyuan chickens.

Gender	Tissue	Correlation Coefficient
Cock	Breast IMP	−0.404
	Leg IMP	0.271
Hen	Breast IMP	0.534
	Leg IMP	0.538

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
