# Peer review of "Correlation Analysis between AK1 mRNA Expression and Inosine Monophosphate Deposition in Jingyuan Chickens"

_animals, 2020, doi:10.3390/ani10030439_

Round 1
Reviewer 1 Report
I have reviewed the manuscript entitled “Correlation analysis between AK1 mRNA expression and inosine monophosphate deposition in Jingyuan chickens” for a publication in Animals as an original article. The authors investigated RNA-seq and real-time PCR expression analyses of breast and leg muscle tissues. Analyses and results look reasonable. However, I think RNA-seq method is not clear. And also, I suggest another statistic analysis. Therefore, I recommend this paper needs to be revised before publishing. My comments will be shown below:
Major comments
- In general, the raw data of RNA-seq should be deposited on SRA in NCBI. Therefore, would you add the accession number in Materials and Methods?
- Methods of RNA-seq were poorly described. Please add more detailed information including sample preparation, kind of NGS platform, and software. If the authors used the data previously reported, it should be mentioned clearly.
- The authors described the relative expression levels in real-time PCR. However, I cannot find what kinds of housekeeping genes the authors used. Please add it.
- Regarding statistics, the authors used one-way ANOVA in order to test the difference between breast and leg muscle in each sex. However, I think that two-way ANOVA should be more suitable in this case. The effects on both muscle part and sex can be tested simultaneously by two-way ANOVA. Although the authors did not focus on, I think sex difference will be matter in the evaluation of the meat quality. Please change to two-way ANOVA and discuss about sex difference.
- In the results of Table 6, there are some differences between male and female. Did the authors make the scatter plot with regression line? By making the scatter plot and testing both muscle part and sex (including interaction-effect) using two-way ANOVA, the authors will find another side of the results. Therefore, I strongly recommend the authors test main-effects and interaction-effect on both muscle part and sex.
Minor comments
- In the first sentence of the Abstract, “In this study,,” should be “In this study,”.
- In the third sentence of the Abstract, “Reaction(PCR)to” should be “Reaction (PCR) to”.
- In the last sentence of 2.3 of Materials and Methods, “Genomes(KEGG)” should be “Genomes (KEGG)”.
- In Table 4, the authors used a red square mark to stress the results. I think this is unusual style for Table. I recommend the authors use both bold letter and superscript number with their explanation in footnote. Please modify it.
- In the second sentence of the third paragraph of Discussion, “Ontology(GO)” should be “Ontology (GO)”.
Author Response
Dear Reviewer:
I hanve modified it according to your requirements.Please see the attachment,kind regards.

Reviewer 2 Report
The manuscript is very informative and maybe can bring some knowledge to the field, which it is inserted. There are some points on english misspelling that should be corrected.
Introduction is clear and very informative; Materials and methods item is current and is giving scientific soundness to the results. The real impact of manuscript is restrict to the country, because it is a chicken line found only in China. The results are well written and comprehensive for all kind of readers. The conclusion could be improved and increased.
Author Response
Dear Reviewer 2,
I have modified it according to your suggestion,kind regards.
(1) In the first sentence of the Abstract, “In this study,,” I have modified “In this study,”.
(2)In the third sentence of the Abstract, “Reaction(PCR)to” I have modified “Reaction (PCR) to”.
(3)In the last sentence of 2.3 of Materials and Methods, “Genomes(KEGG)” I have modified “Genomes (KEGG)".
(4)In the second sentence of the third paragraph of Discussion, “Ontology(GO)” I have modified “Ontology (GO)”.
(5)The conclusion I have modified: "To the best of our knowledge, in this study, we discussed the differences of IMP mechanism in Jingyuan chicken due to different parts and sexes. through real-time quantitative PCR and correlation analysis, that the accuracy of screening differentially expressed genes by transcriptome sequencing is high. Successfully screened Ak1,and AK1 promoted the formation of inosine acid in Jingyuan chicken; but the specific function and mechanism of Ak1 is unclear. Therefore, it is necessary to conduct big data association analysis and functional research in the future. Screening of the Jingyuan chicken inosinic acid synthesis-related gene can be used as a marker for future marker-assisted selection in breeding".
Round 2
Reviewer 1 Report
I think the authors revised well.
This manuscript is a resubmission of an earlier submission. The following is a list of the peer review reports and author responses from that submission.